# Assessment of the Potential Suitable Habitat of *Apriona rugicollis* Chevrolat, 1852 (Coleoptera: Cerambycidae) Under Climate Change and Human Activities Based on the Biomod2 Ensemble Model

**DOI:** 10.3390/insects15120930

**Published:** 2024-11-27

**Authors:** Liang Zhang, Chaokun Yang, Guanglin Xie, Ping Wang, Wenkai Wang

**Affiliations:** 1Institute of Entomology, College of Agriculture, Yangtze University, Jingzhou 434025, China; 13349832924@163.com (L.Z.); yang1764847081@163.com (C.Y.); xieguanglin@yangtzeu.edu.cn (G.X.); 2MARA Key Laboratory of Sustainable Crop Production in the Middle Reaches of the Yangtze River (Co-Construction by Ministry and Province), College of Agriculture, Yangtze University, Jingzhou 434025, China

**Keywords:** *Apriona rugicollis*, species distribution model, ensemble model, climate change, human activities

## Abstract

*Apriona rugicollis* Chevrolat, 1852 is an important phytophagous pest that mainly affects plants of the Moraceae, Salicaceae, and Ulmaceae families. In this study, an ensemble model was used to predict the potential habitats of *A. rugicollis* and their areas of change under current and future climate change conditions. The results show that the habitat of *A. rugicollis* will gradually expand to northern China and Hokkaido, Japan, in the future with global climate change, suggesting that it is necessary to establish an early warning and monitoring network in these regions to prevent its further spread and thus potential damage to host plants. This study provides a theoretical and scientific basis for the prevention and control of the spread of *A. rugicollis*.

## 1. Introduction

Increased greenhouse gas emissions resulting from intensified human activities have significantly altered the global climate [1]. Global warming events not only affect global temperatures and precipitation patterns but also have a profound impact on the ranges and habitats of species in ecosystems, with a particularly pronounced effect on insects [2]. As important components of ecosystems, insects play an integral role in maintaining ecological balance, promoting biodiversity, and facilitating ecological processes [3]. As climate change intensifies, the habitats of many insects are changing, posing a potential threat to ecological balance [4]. Global warming has led to an increase in the frequency of extreme weather, accelerating the movement and expansion of insect habitats, and some insects have begun to migrate to higher latitudes or higher altitudes in search of suitable living environments [5]. This migration not only affects the survival and reproduction of the insects themselves but may also disrupt their interrelationships with host plants, which in turn affects the health of the entire forest ecosystem [6,7]. In addition, although relatively few studies have been conducted on the effects of human activities on the distribution ranges of insects, it has been shown that human activities, especially deforestation, urbanization, and agricultural development, indirectly affect the distribution patterns of insect populations through a variety of ways, including changes in habitat structure, food resource availability, and habitat connectivity [8,9,10]. In order to effectively address these challenges, it is important to enhance the prediction of potential distribution areas of pests and develop adaptation and mitigation strategies based on the prediction results. These strategies will provide scientific support for ecosystem management and sustainable development and help to better protect and utilize forest resources.

Species distribution models (SDMs), as an important modeling tool in ecology and biogeography, are effective in predicting the potential distribution of species under different environmental conditions [11]. Correlative species distribution models (SDMs) simulate suitable habitats for species by combining statistical and machine learning algorithms using their occurrence records and environmental variables and are irreplaceable in assessing the potential impacts of climate change on species distribution [12,13]. SDMs help ecologists and policymakers gain a deeper understanding of species’ responses to climate change, thus providing a scientific basis for developing effective conservation measures [14]. Ensemble models (EM) are considered to be more reliable and powerful for predicting species distribution in SDMs. Compared with a single model, EM can predict results more accurately by integrating the outputs of multiple models [15]. EM is widely used in biodiversity conservation planning, invasive species management, assessing the impacts of climate change, and pest monitoring [16]. Especially in pest management, it can provide a scientific basis for managers to carry out monitoring and control by predicting the potential distribution areas of pests, which not only helps to formulate targeted monitoring and management strategies but also significantly improves preventive measures [17,18]. By applying EM, we can more effectively protect ecosystems and forest production and reduce the negative impacts of pests.

*Apriona rugicollis* Chevrolat, 1852 (Coleoptera, Cerambycidae, Lamiinae) is an important polyphagous longhorn beetle, widely distributed in North Korea, South Korea, Japan, and China in East Asia and mainly infesting Moraceae, Salicaceae, and Ulmaceae, including economically important species grown for fruit, timber, bonsai cultivation, and silkworm cultivation (mulberry) [19]. Larval feeding activity may hollow out smaller branches, causing die-back and collapse, and attack by multiple larvae can weaken and kill entire trees and increase their susceptibility to windbreak [20]. Attacked mulberry trees become stunted, and fig trees die-back and fail to fruit [21]. Adults feeding on the bark or leaves of shoots can cause the lower sapwood to die or even trigger the drying out of the entire plant, producing an infestation that may be localized over a large area, seriously affecting the health and growth of the tree [22]. *A*. *rugicollis* not only has a severe impact on tree growth but also causes significant economic and ecological losses. Although no specific economic losses have been explicitly reported in the literature, the area of forest damaged by the feeding behavior of *A*. *rugicollis* continues to expand [23]. Its spread not only exacerbates local ecological degradation but also poses a serious challenge to the sustainable management of forest resources [24]. Therefore, understanding the potential habitat and distribution of *A. rugicollis* under future climate scenarios will help decision-makers conduct monitoring, prevention, and management programs within forest production areas to safeguard the health of forest production.

Understanding the habitat suitability of pests under future climate scenarios is essential for ensemble pest management. In this study, we used an optimized ensemble model to analyze potential areas of relative change in the geographic distribution of *A. rugicollis* under different current and future climate conditions. The main objectives of the study included the following: (1) identifying key environmental variables affecting the distribution of *A. rugicollis*; (2) comparing differences in the extent and area of the distribution of *A. rugicollis* habitat in the current period under scenarios with and without anthropogenic activities; (3) predicting future changes in the potential geographic distribution area of *A. rugicollis* and its area under different climate change scenarios; and (4) analyzing the spatial distribution dynamics of *A. rugicollis* and its future development trend. The results of the study reveal the far-reaching impacts of climate change on the distribution of *A. rugicollis* and provide theoretical support and a practical basis for the scientific formulation of management strategies and countermeasures.

## 2. Materials and Methods

### 2.1. Species Occurrence Data

To construct *A. rugicollis* occurrence data for use in Biomod2 modeling, we collected data from multiple reliable sources. These sources include (1) literature and online references (CNKI, https://www.cnki.net, accessed on 1 October 2024; WOS, and https://www.webofscience.com/wos; NACRC, http://museum.ioz.ac.cn, accessed on 1 October 2024; (2) Global Biodiversity Information Facility (GBIF) (https://doi.org/10.15468/dl.kb7hbz, accessed on 2 October 2024); (3) iNaturalist (https://www.inaturalist.org, accessed on 1 October 2024); (4) European and Mediterranean Plant Protection Organization (EPPO) Database (https://gd.eppo.int/taxon/APRIJA/distribution, accessed on 30 September 2024); and (5) data from field surveys conducted by researchers from 2013 to the present in various provinces in China. For records lacking specific latitude and longitude information, we used Google Earth software (http://ditu.google.cn, accessed on 8 October 2024) to obtain the geographic centroid of the corresponding administrative division as the approximate latitude and longitude coordinates based on its geographic description. With these diverse data sources, we collected a total of 1017 distribution points (Figure 1), which provided sufficient data support for the construction of the ensemble model.

To avoid overfitting of the model input data, we first removed duplicate occurrences and then sparsified the data using the “spThin” package (version 0.2.0) in R to ensure the quality of the dataset and minimize potential bias caused by spatial clustering, which improves the prediction accuracy and robustness of the model [25]. The sparsification distance was aligned with the size of the environmental factor grid cells, which was used to eliminate redundant data recorded multiple times in the same grid, thus further optimizing model performance. Ultimately, we retained 434 occurrences of *A. rugicollis* for the construction of the ensemble model.

### 2.2. Environmental Variables Data

In this study, in order to analyze and predict the potential current and future distribution areas of *A. rugicollis*, we downloaded 19 bioclimatic variables at a resolution of 2.5 arc-minutes (4.6 km) from the WorldClim climate database (https://www.worldclim.org, accessed on 20 December 2023). Among them, current climate conditions were based on observational records from 1970–2000, while future climate data were adopted from the Beijing Climate Center Climate System Model (BCC-CSM2-MR) of the Sixth International Coupled Model Intercomparison Project (CMIP6) [26]. For future climate data from 2041–2060 (2050s) and 2061–2080 (2070s), four shared socioeconomic pathway (SSP) scenarios were selected for the future scenario simulations, namely the low forcing scenario (SSP1-2.6), the medium forcing scenario (SSP2-4.5), the medium-high forcing scenario (SSP3-7.0), and the high forcing scenario (SSP5-8.5). These scenarios are based on different socio-economic development paths and reflect the future socioeconomic development plans of each country [27]. Meanwhile, elevation data for 2.5 arc-minutes were downloaded from WorldClim, and slope and aspect data were extracted using ArcGIS Map (version 10.8.1) software (WGS 1984). We also downloaded solar radiation data from the Helmholtz Center for Environmental Research (https://www.ufz.de/gluv, accessed on 20 January 2024). In addition, we downloaded human activity data from the Socioeconomic Data and Applications Center (https://sedac.ciesin.columbia.edu, accessed on 15 January 2024), including the global human impact index, the global human footprint, and global population density. World and administrative division maps were obtained from the standard map service website of the National Bureau of Surveying, Mapping and Geographic Information (http://bzdt.ch.mnr.gov.cn/index.html, accessed on 9 December 2023). Finally, we used the “Resample” and “Extract” tools in the ArcGIS Map software to normalize the 31 environmental variables into a uniform format for subsequent analysis and modeling.

The selection and treatment of environmental variables are extremely critical when constructing species distribution models because high correlations between variables may trigger autocorrelation and multicollinearity problems, which can have an impact on the accuracy of model predictions [28,29]. To address these issues, we assessed the correlations among 31 environmental variables and used the variance inflation factor (VIF) to check for multicollinearity. Following Mulatu et al. [30] approach, we first performed a Pearson correlation analysis (Figure 2) on the 31 bioclimatic variables using the “car” package (version 3.1.2) in the R (version 4.4.1) software and selected the variables with correlation coefficients |*r*| ≤ 0.7 for modeling. When the correlation coefficient |*r*| > 0.7 between two variables, the “usdm” package (version 2.1.7) was used to exclude variables with high VIF value (VIF > 10) to avoid multicollinearity problems. Finally, eight environmental factors (Table 1) were selected that were relatively independent and had a large impact on model predictions, and these variables ensured the accuracy and reliability of the model.

### 2.3. Algorithms, Construction, and Validation of Ensemble Models

We conducted a habitat suitability analysis of *A. rugicollis* using the “Biomod2” package (version 4.2.5), combining the predictions of 11 different models through an ensemble modeling approach to improve the predictive accuracy of the models in a weighted average manner (Wmean) [31]. Models provided by Biomod2 include the Generalized Linear Model (GLM), the Generalized Additive Model (GAM), Multivariate Adaptive Regression Spline (MARS), Flexible Discriminant Analysis (FDA), Random Forest (RF), Maximum Entropy Model (MAXENT), Generalized Boosting Model (GBM), Classification Tree Analysis (CTA), Artificial Neural Networks (ANNs), Surface Range Envelope of the Profile Model (SRE) and Extreme Gradient Boosting (XGBOOST) [32].

In this study, we used AUC (Area Under the Receiver Operating Characteristic Curve) and TSS values (True Skill Statistic) as the core metrics for assessing and validating the accuracy of the model [33,34]. The AUC value is one of the most important measures of model accuracy and predictive performance and is used to assess the overall performance of a model by quantifying its classification ability [35]. The value of AUC ranges from 0 to 1, and the larger the AUC value, the better the prediction performance of the model, indicating that the model is more effective in distinguishing between the actual distribution and non-distribution areas of the species. Although AUC values are widely used in SDMs, they are not the only criteria for assessment [36]. The TSS value is a statistic based on the confusion matrix that combines the sensitivity (quantification of omission error) and specificity (quantification of commission error) of the model. TSS values range from −1 to 1, with values closer to 1 indicating a better predictive ability of the model, while values closer to 0 or negative values indicate a poorer predictive effect of the model. By combining the two complementary metrics, AUC and TSS value, we are able to assess the reliability and validity of the model in a more comprehensive manner, thus further deepening our understanding of the model’s predictive performance.

We first performed a preliminary analysis of these 11 models, selecting only those with AUC values greater than 0.9 and TSS values greater than 0.7 as the base models for final modeling. The modeling dataset was divided into 80% species occurrence records as a training set and 20% as a test set, in addition to 1000 randomly generated pseudo-absence points. In order to enhance the confidence of the models, 10 repetitions of each model were performed, and a k-fold cross-validation method was used to reduce model variance and improve consistency across different data subsets. At the same time, the model parameters were optimized using the auto-tuning strategy (tuned) provided by the Biomod2 package to obtain better predictive performance. Finally, based on the best-performing models, we calculated the contribution of each environmental variable in the ensemble model and created response curves of species survival probabilities to key variables to gain a comprehensive understanding of the impact of these variables on *A. rugicollis* distributions.

### 2.4. Analyzing the Potential Distribution of A. rugicollis Under Current and Future Scenarios

We investigated the effects of natural environmental variables and human activities on the geospatial distribution pattern of *A. rugicollis* by building two different ensemble models: (1) prediction using only natural environmental factors (including current bioclimate + topography + solar radiation) for the current period; (2) predictions using current natural environmental factors and human activity data (including global human footprint, global human impact index, and population density). In addition, on the basis of model (1), the area of distribution at a future time is predicted (future extension of model 1) by combining future natural environmental factors (including future bioclimatic + topographic + solar radiation). Models (1) and (2) were based on current climate patterns to assess anthropogenic impacts on the habitat suitability of *A. rugicollis*. In contrast, model (1) and its future extensions explore the potential impacts of climate change on the habitat suitability of *A. rugicollis* by comparing current and future climate patterns. These models reveal the relative importance of natural factors and human activities in species distribution.

The predictions generated by the ensemble models are presented as a continuous ASCII raster layer, where each pixel represents a probability of presence (*P*) value for *A. rugicollis*, ranging from 0 to 1000. We generate binary maps based on the specificity threshold of the TSS value by classifying regions with presence probability values below the threshold as unsuitable and regions above the threshold as suitable [37]. This classification can visualize the suitable and unsuitable areas for *A. rugicollis*, which can help to further analyze the distribution pattern of *A. rugicollis* and its relationship with environmental factors.

### 2.5. Spatial Distribution Dynamics of A. rugicollis and Movement Paths of Potential Geographic Distribution Centers Under Different Future Climate Scenarios

We used the Biomod2 package to assess changes in the distribution of *A. rugicollis* under different future climate change scenarios by analyzing binomial maps under different future climate change scenarios in comparison with those under current climate scenarios. The results of the changes were categorized into four groups: “Expansion” (species range increase), “No occupancy” (no distribution in both current and in the future), “No change” (species range remains unchanged), and “Shrinking” (species range decreases). We also used the “Centroid Changes (Lines)” tool in the “SDMToolbox” package (version 2.6) to identify potential centers of geographic distribution of *A. rugicollis* at different times in the current and future [38]. By linking the centers of distribution in different time periods and under different carbon emission scenarios, the spatial trajectories of suitable habitats for *A. rugicollis* under future climate conditions were mapped. This analysis not only reveals the response patterns of species to climate change but also provides an important spatial reference for predicting future changes in the spatial distribution of species.

## 3. Results

### 3.1. Accuracy of the Ensemble Model

In this study, the accuracy of the ensemble models in predicting the distribution of *A. rugicollis* was assessed by AUC values and TSS values. The models were run 10 times, and we obtained average TSS and AUC values for the 11 models. The results show that when using only natural environment factors, ANN, GAM, and SRE were excluded from the ensemble model due to their low TSS and AUC values, and we chose RF, CBM, CTA, XGBOOST, MAXENT, MARS, FDA, and GLM to construct the ensemble model. When adding human activity variables, we chose ANN, RF, CBM, CTA, XGBOOST, MAXENT, MARS, FDA, and GLM to construct the ensemble model (Figure 3).

When considering only the natural environment factor, the ensemble model had an AUC value of 0.99 and a TSS value of 0.89. However, after adding the human activity variable, the AUC value was 0.98, and the TSS value was 0.87 (Table 2). This demonstrates the high predictive ability and accuracy of the two ensemble models to effectively predict and validate the distribution of *A. rugicollis* and, likewise, further demonstrates the generalization ability and stability of the models.

### 3.2. Assessment of Environmental Variables

The results of the study show that the weights of the eight environmental variables varied across SDMs. Specifically, when only natural environmental factors were considered, precipitation seasonality (Bio15, 8.2%), precipitation of the coldest quarter (Bio19, 48.9%), and UV-B_seasonality (Bio24, 37.0%) had a significant effect on the distribution of *A. rugicollis,* with a cumulative contribution of 94.1%. The main variables affecting the potential geographic distribution of *A. rugicollis* under the combined effect of natural environmental factors and human activities were the precipitation of the coldest quarter (Bio19, 12.8%), UV-B_seasonality (Bio24, 33.1%), global human footprint (Bio29, 16.9%), and population density (Bio31, 30.4%), with a cumulative contribution of 93.2%. Despite the variation in the contribution of each environmental factor, Bio19 and Bio24 remained the most dominant natural environmental factors. It is noteworthy that the addition of human activities decreased the contributions of Bio15, Bio19, and Bio24 by 7.4%, 36.1%, and 3.9%, respectively, while the contribution of Bio22 increased by 3.6% (Figure 4). This variation suggests that bioclimate, solar radiation, and human activities together influence the suitability of *A. rugicollis* distribution.

In addition, Figure 5 illustrates the response curve of the distribution probability of *A. rugicollis*, revealing the optimum range of natural environmental factors and human activities on its distribution. The response curves showed that the distribution probability of *A. rugicollis* increased sharply to the highest point with the increase in Bio19 (Bio19 = 264.1818 mm; *P* = 0.8629) and then gradually and slowly declined after it entered into the optimum survival range. Bio24 reached its optimum value at 134,541.7 with a survival probability (*P*) of 0.8276, indicating that bioclimatic factors and solar radiation are key factors in maintaining the distribution of *A. rugicollis*. Similarly, the response curves of Bio29 and Bio31 showed that the distribution probability of the species gradually increased as the intensity of human activities increased, indicating that human activities showed a positive correlation in affecting its geospatial distribution, and these results together reveal that the natural environment and human activities are key factors influencing the distribution of *A. rugicollis*.

### 3.3. Habitat Suitability for the Current Period

The area and region of distribution of *A. rugicollis* in the current period were predicted using an ensemble model (Figure 6). The results show that *A. rugicollis* was mainly distributed in East Asia, mainly concentrated in North Korea, South Korea, Japan, Myanmar, Vietnam, and China. The model prediction results are highly consistent with the actual observed distribution, which again verified the accuracy and practicality of the model. The total area of suitable habitat for *A. rugicollis* was 255.58 × 10^4^ km^2^ when only natural environmental factors were considered (Table 3). However, when the effects of human activities were further included, the total area of suitable habitat was 243.75 × 10^4^ km^2^, a decrease of 11.83 × 10^4^ km^2^, and the areas of decrease were mainly located in Kachin State, Myanmar, Guangxi, Guangdong, Fujian, Jiangxi, Hunan, Hubei, Shandong, and Henan, China, which indicating that the increase in human activities not only changed the distribution pattern of *A. rugicollis* but also reduced the distribution pattern of *A. rugicollis*.

### 3.4. Habitat Suitability for Different Future Climate Scenarios

Figure 7 shows the potential distribution areas of *A. rugicollis* under four different shared socioeconomic pathways (SSP1-2.6, SSP2-4.5, SSP3-7.0, and SSP5-8.5) and two time periods (2050s and 2070s). The results show that the suitable habitat areas for *A. rugicollis* under future climate conditions were generally consistent with those predicted under current climate conditions and remained concentrated in North Korea, South Korea, Japan, Myanmar, Vietnam, and China. Furthermore, the total suitable habitat area for *A. rugicollis* under future climate conditions increased compared with the current period, which suggests that there may be a tendency to expand its potential range in the future. Specifically, the total area of suitable habitat for *A. rugicollis* in the future is predicted to range from 319.93 to 390.10 × 10^4^ km^2^ (Table 3). Among these, the largest area of suitable habitat is predicted for the SSP5-8.5-2070s, followed by SSP3-7.0-2070s, and the smallest area of suitable habitat for the SSP1-2.6-2050s scenario. It is worth noting that under the four future climate scenarios, the total suitable habitat area of *A. rugicollis* showed a positive trend of increasing over time. This expansion trend reflects that future climate change may provide more potentially suitable areas for habitat expansion of *A. rugicollis*, thus affecting the distribution dynamics and ecological adaptations of the species.

### 3.5. Trends in the Development of Suitable Habitat for A. rugicollis Under Future Climate Change Scenarios

We assessed the relative change in the area and region of suitable habitat for *A. rugicollis* based on the differences in potentially suitable areas under future climate scenarios. The results show that under future climate scenarios, the suitable growing areas of *A. rugicollis* are expanding to Hokkaido in Japan, Duanchon and Gangjeok in North Korea, Pyeongchang and Gangwon-do in South Korea, and Beijing, Hebei, Shandong, Zhejiang, Anhui, Hunan, Gansu, Shaanxi, Henan, Shanxi, Fujian, Liaoning, Jilin, Heilongjiang, Sichuan, and Yunnan in China (Figure 8), with expansion areas ranging from 71.68 to 140.78 × 10^4^ km^2^ (Table 4). Among them, the SSP5-8.5-2070s scenario had the largest expansion area, followed by SSP3-7.0-2070s, while SSP1-2.6-2050s had the smallest expansion area. Meanwhile, the contraction areas of suitable habitats were mainly located in Jiangxi, Taiwan, Hainan, Henan, Anhui, and Xizang in China, Kachin State in Myanmar, and Nghe An and Ha Tinh in Vietnam, with contraction areas ranging from 3.33 to 9.75 × 10^4^ km^2^. Among them, the SSP3-7.0-2070s scenario has the largest contraction area, followed by the 1-2.6-2050s, while the SSP3-7.0-2050s has the smallest contraction area.

### 3.6. Changes in Potential Distribution Centers of A. rugicollis Under Future Climate Change Scenarios

The shift in the potential distribution center of *A. rugicollis* reveals a tendency to migrate to higher latitudes under different climate scenarios in the future, with an overall shift to the northeast (Figure 9). In the current period, the distribution center of *A. rugicollis* was located in Shiyan City, Hubei Province, China (32.43° N, 110.06° E). In the SSP1-2.6 scenario, the center of distribution in the 2050s was located in Binzhou City, Shandong Province, China (38.16° N, 118.11° E), and moved to Jinghai District, Tianjin City, China (117.04° N, 38.77° E), in the 2070s. In the SSP2-4.5 scenario, the potential distribution center was located in Jinnan District, Tianjin City (117.28° N, 38.94° E), in the 2050s and then moved northeastward by about 114.96 km to reach Qinhuangdao City, Hebei Province (119.15° N, 40.09° E), in the 2070s. Under the SSP3-7.0 scenario, the distribution centers were located in Jinnan District, Tianjin City (117.23° N, 38.94° E), and Tangshan City, Hebei Province (119.28° N, 40.31° E), in the 2050s and 2070s, respectively. Under the SSP5-8.5 scenario, the distribution center of *A. rugicollis* was located in Xiqing District, Tianjin City (117.05° N, 39.04° E), in the 2050s, and then moved to Chengde City, Hebei Province (117.79° N, 40.80° E), in the 2070s (Table 5). This change in movement suggests that, over time, *A. rugicollis* has migrated to higher latitudes in response to climate change in order to secure suitable environments for its survival and reproduction. This behavior also reflects an adaptive response of *A. rugicollis* to environmental change, aiming to find more suitable habitats and thus improve its survival.

## 4. Discussion

Understanding the environmental variables that affect suitable habitats for pests is critical to developing effective monitoring and control programs [39]. By identifying these key environmental factors using species distribution models, we can more accurately predict the potential range of pests and thus develop targeted control measures to prevent their spread and invasion [40,41]. This will not only help protect ecosystems and agroforestry resources but is also critical for maintaining biodiversity [24,42]. Based on the above, this study used an ensemble model to predict the potential geographic distribution of *A. rugicollis* and its areas of change under current and future climate change. The results of this study provide an important reference for the global forestry economy and phytosanitary work, and not only provide a scientific basis for governments to formulate scientific pest control and management measures and carry out ecological protection work but also effectively respond to the potential threat of climate change to forestry production, which is of great economic and ecological significance, and contributes to the realization of the goal of sustainable development.

### 4.1. Reliability and Accuracy of Ensemble Model

Compared with a single SDM, the ensemble modeling approach effectively reduces the limitations of a single model and results in more robust and accurate predictions [43,44]. Especially in complex ecosystems or where data are unevenly distributed, the ensemble approach demonstrates greater adaptive capacity [17,45,46]. In this study, we used multiple models for modeling and significantly improved the accuracy of the ensemble models by optimizing the model parameters. The accuracy evaluation metrics of our two created models (including the model of anthropogenic factors and the model of non-anthropogenic factors) reached a high standard. The TSS value of the model exceeds 0.87, indicating that the model can accurately distinguish between suitable and unsuitable areas while maintaining a low misclassification rate [33,35]. Meanwhile, the AUC value is greater than 0.98, which further proves the accuracy of the model, reflecting a high true positive rate and a low false positive rate. These results indicate that the ensemble model we created has strong reliability and accuracy in predicting the potential distribution of *A. rugicollis*, which provides scientific support and a basis for developing effective monitoring, control, and management strategies.

### 4.2. Analysis of Environmental Variables

Insect survival probabilities are closely linked to environmental variables, and precipitation, solar radiation, and human activities especially have become important drivers of insect distribution and survival [47]. These variables not only have an impact on the life cycle and reproduction rate of insects but also influence their habitat selection, which in turn directly or indirectly determines their distributional range [48].

The response curve showed that the survival probability of *A. rugicollis* gradually increased with the increase in precipitation of the coldest quarter (Bio19). However, when precipitation exceeds its tolerance limit (264.1818 mm), it may adversely affect its physiological activities and thus threaten its survival. In order to respond to excess precipitation, *A. rugicollis* may tend to seek drier habitats, which could affect its choice of mating and egg-laying sites and further alter the range of the population [49,50].

Solar radiation, as an important source of energy, also has a significant effect on the development and reproduction of *A. rugicollis*, which in turn alters its activity rhythms [51]. Low-intensity UV-B_seasonality (Bio24) may stimulate the initiation of physiological mechanisms in *A. rugicollis*, thereby increasing the probability of insect survival. At the same time, low-intensity radiation may also serve as an environmental signal that prompts *A. rugicollis* to adjust its physiological and behavioral states to better adapt to environmental changes, further improving its ability to survive in the environment. However, high-intensity UV-B_seasonal (Bio24) may cause serious harm to insects [52]. Additionally, excessive radiation generates too many reactive oxygen radicals, which exceed the scavenging capacity of the antioxidant defense system in insects, leading to severe oxidative damage to cells [53]. Furthermore, high-intensity radiation may directly damage the insect’s nervous system and endocrine system, interfering with its physiological regulation and hindering life activities such as metabolism and reproduction, leading to a decrease in the probability of survival.

In addition, with the increase in human afforestation activities, especially the creation of plantation forests of mulberry plants, more habitats are provided for *A. rugicollis* [54]. At the same time, human activities may lead to a decrease in the number of natural enemies of *A. rugicollis*, which, in a way, also creates favorable conditions for the survival of *A. rugicollis* [1,55].

### 4.3. Potential Distribution Areas and Displacement

The Sixth Assessment Report (AR6) of the United Nations Intergovernmental Panel on Climate Change (IPCC) states that there is a greater than 50% chance that global temperatures will increase by 1.5 °C between 2021 and 2040, and that by 2100, global temperatures could rise by 3.3 °C to 5.7 °C [56]. It is worth noting that the last time global temperatures exceeded pre-industrial levels was more than 3 million years ago [4,57]. Global warming may be one of the key drivers of changes in suitable habitats for *A. rugicollis*. According to the simulation results of the ensemble model, under the future climate scenarios, the suitable habitat of *A. rugicollis* shows significant spatial variation, and its suitable growing area is expanding to Hokkaido, Japan, Duanchon and Gangjae, North Korea, Pyeongchang-gun, South Korea, and northern China. Although the expansion trend varied in different future scenarios, the area of expanding areas was always larger than that of contracting areas. This change reveals the far-reaching impact of climate change on insect distribution and also provides an important reference for decision-makers in ecological management and pest monitoring.

In recent years, many insects have gradually migrated to higher latitudes, a phenomenon closely linked to climate change [58]. Increased temperatures lead to changes in the habitat of insects, thus affecting their survival and distribution [59]. The present study also proves this conclusion again: the center of mass of *A. rugicollis* was shifted to higher latitudes, and the overall trend was to the northeast. The migration of *A. rugicollis* not only affects its survival and dispersal but also has significant impacts on ecosystems, forestry production, and biodiversity conservation of the new distribution area. As *A. rugicollis* expands into new suitable habitats, ecosystem composition and functioning may change, leading to changes in the underlying structure of the food chain and ecological balance. Therefore, management strategies for *A. rugicollis* need to take into account potential future distributional changes in order to meet the challenges posed by the new distributional patterns of the species and to ensure ecological and economic security in the new range.

### 4.4. Limitations of This Study and Future Research Directions

In this study, we used ensemble modeling to reveal the potential effects of climate change on the distribution of *A. rugicollis*. Although we obtained some important findings, there are still several limitations in this study. (1) The unicity of climate models is an important limitation of this study. Although the BCC-CSM2-MR model provides valuable predictions of future climate change, the use of a single climate model may lead to biased predictions [60]. Future studies should consider the use of different global climate models (GCMs) that can contribute to a more comprehensive understanding of the potential impacts of climate change on *A. rugicollis*. (2) The environmental factors in this study relied heavily on modeling of bioclimate, topography, solar radiation, and human activities and failed to adequately consider the important impacts of other environmental factors (e.g., soil and vegetation type) on species distribution [61]. Future studies should integrate more ecological and environmental factors to improve the accuracy and usefulness of the model. (3) The present study considered the pest (*A. rugicollis*) singularly and neglected the complex interactions with its host plants and natural enemies. Future studies should include host plants and natural enemies in the same framework to explore their interrelationships and mechanisms of influence. (4) As *A. rugicollis* expands into new suitable habitats, it may have significant impacts on ecosystems, forestry production, and biodiversity conservation in its new range [62]. Therefore, ecological risk assessment and management recommendations for organisms in the predicted potential distribution areas are important components of future research.

In conclusion, this study provides a preliminary understanding of the changes in suitable habitats for *A. rugicollis* in East Asia. Despite some limitations, future studies can build on this foundation and further explore the relationship between climate change and ecological management. This will help to develop more effective monitoring and management strategies to address the potential impacts of climate change on ecosystems.

## 5. Conclusions

In this study, an ensemble model was applied for the first time to identify the key environmental factors affecting the distribution of *A. rugicollis* and to predict its potential habitat and relative change areas under current and future climatic conditions. The results show that humidity, solar radiation, topography, and human activities were the main factors affecting the distribution of *A. rugicollis*. Under current climate scenarios, suitable habitats are concentrated in East Asia, including North Korea, South Korea, Japan, Myanmar, Vietnam, and China. Under different future climate scenarios, the area of suitable habitats for *A. rugicollis* gradually increases in China and Japan, suggesting that these regions will be at higher risk of dispersal. In addition, the ensemble model predicts that suitable habitats for *A. rugicollis* will expand northeastward to higher latitudes. Therefore, future work should strengthen preventive and control measures in northern China and Hokkaido, Japan, and an early warning and monitoring network needs to be established in advance to prevent the potential damage to host plants caused by the further spread of *A. rugicollis*. The results of this study provide an important scientific basis for decision-makers to develop monitoring and control measures for *A. rugicollis* under future climatic conditions.

## Figures and Tables

**Figure 1 insects-15-00930-f001:**
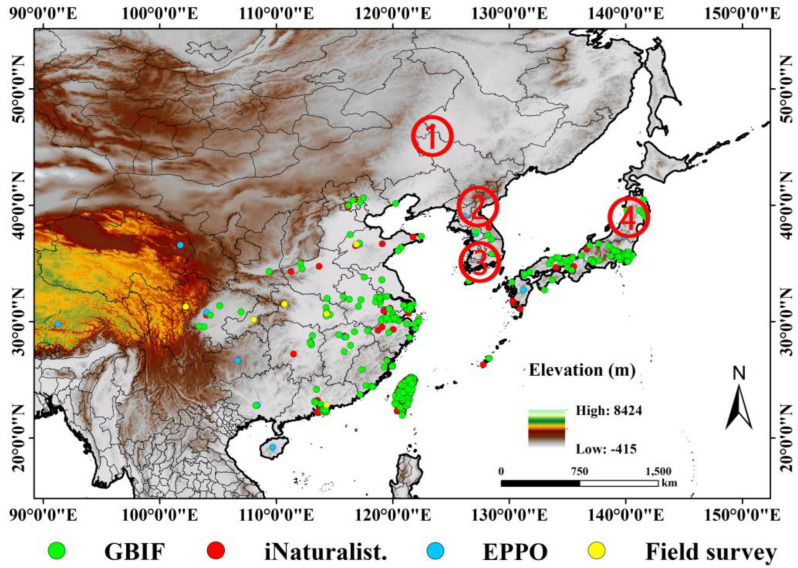
Occurrence records of *A. rugicollis*: ① China; ② North Korea; ③ South Korea; ④ Japan.

**Figure 2 insects-15-00930-f002:**
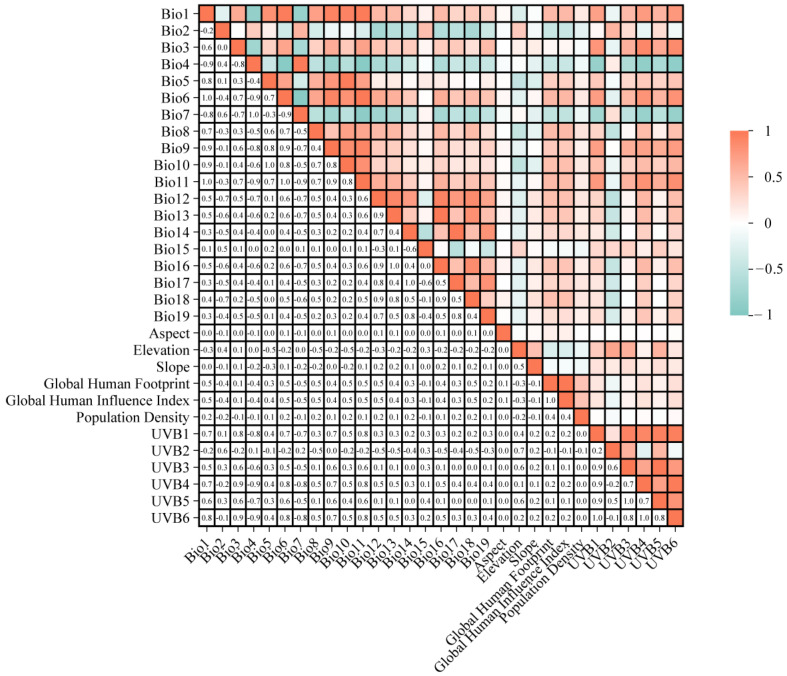
Correlation among the 31 environmental variables.

**Figure 3 insects-15-00930-f003:**
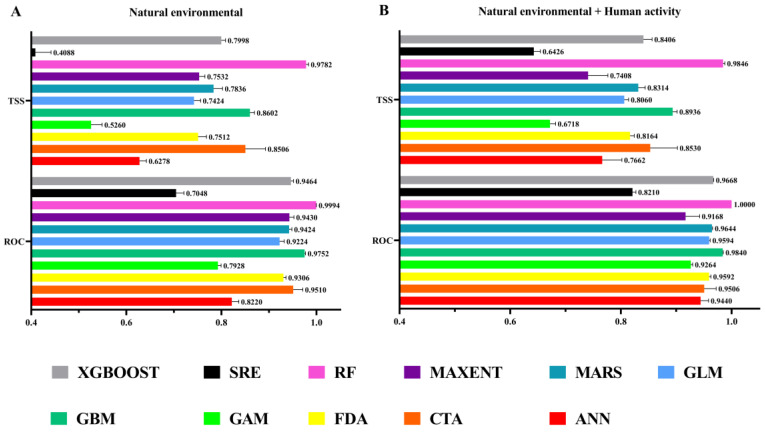
Accuracy evaluation of eleven single models: (**A**) natural environmental; (**B**) natural environmental + human activity.

**Figure 4 insects-15-00930-f004:**
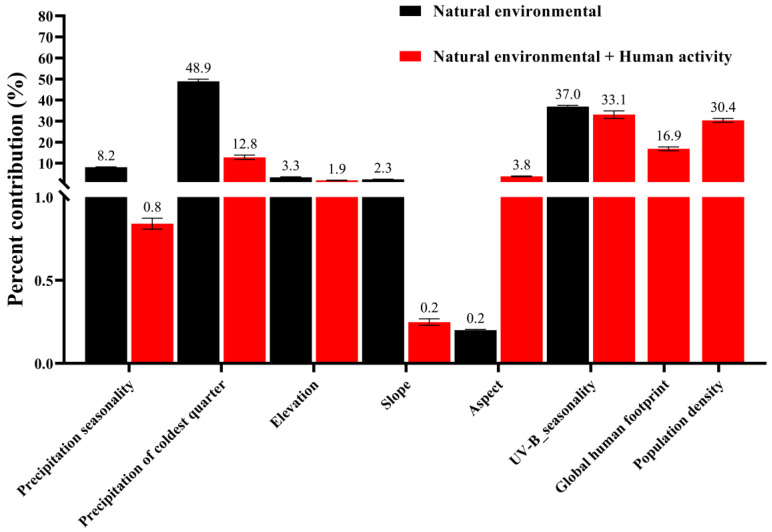
Importance of the environmental factors.

**Figure 5 insects-15-00930-f005:**
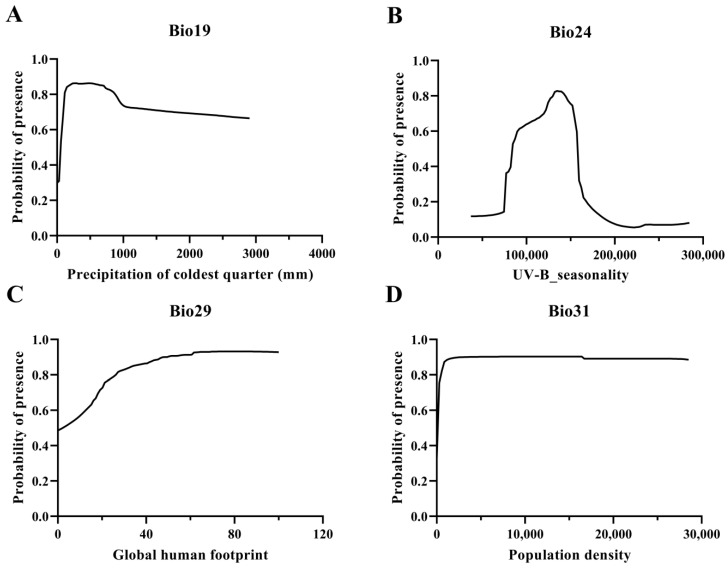
Response curves for environmental factors: (**A**) Bio19; (**B**) Bio24; (**C**) Bio29; (**D**) Bio31.

**Figure 6 insects-15-00930-f006:**
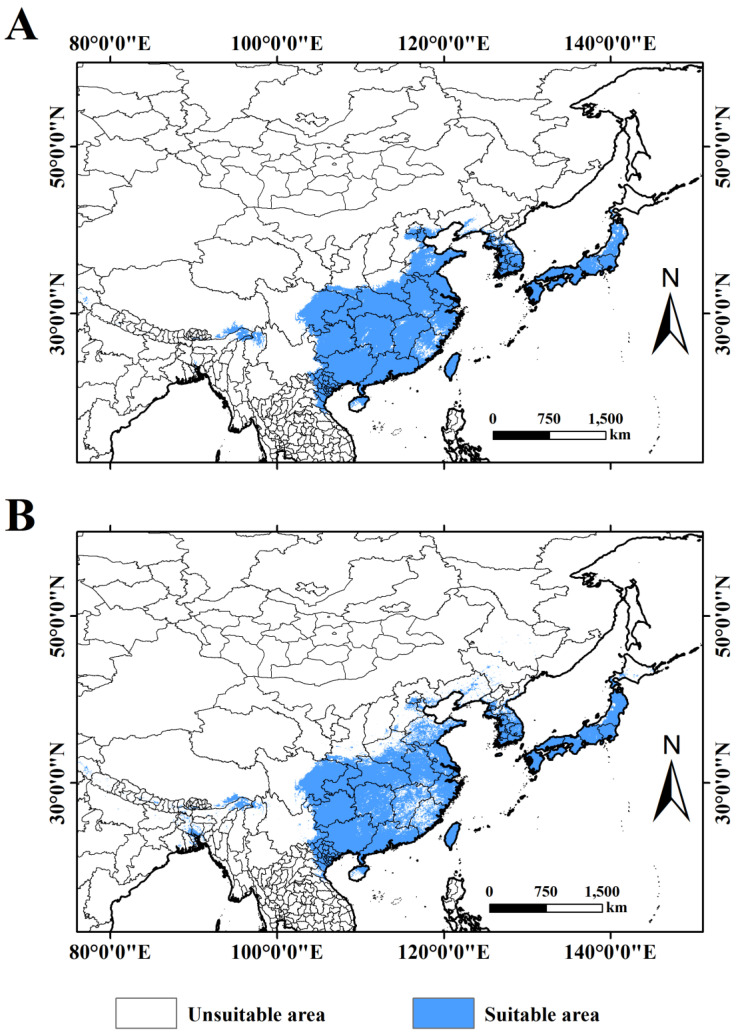
Predicted distribution areas of *A. rugicolli* under current climate models: (**A**) natural environmental; (**B**) natural environmental + human activity.

**Figure 7 insects-15-00930-f007:**
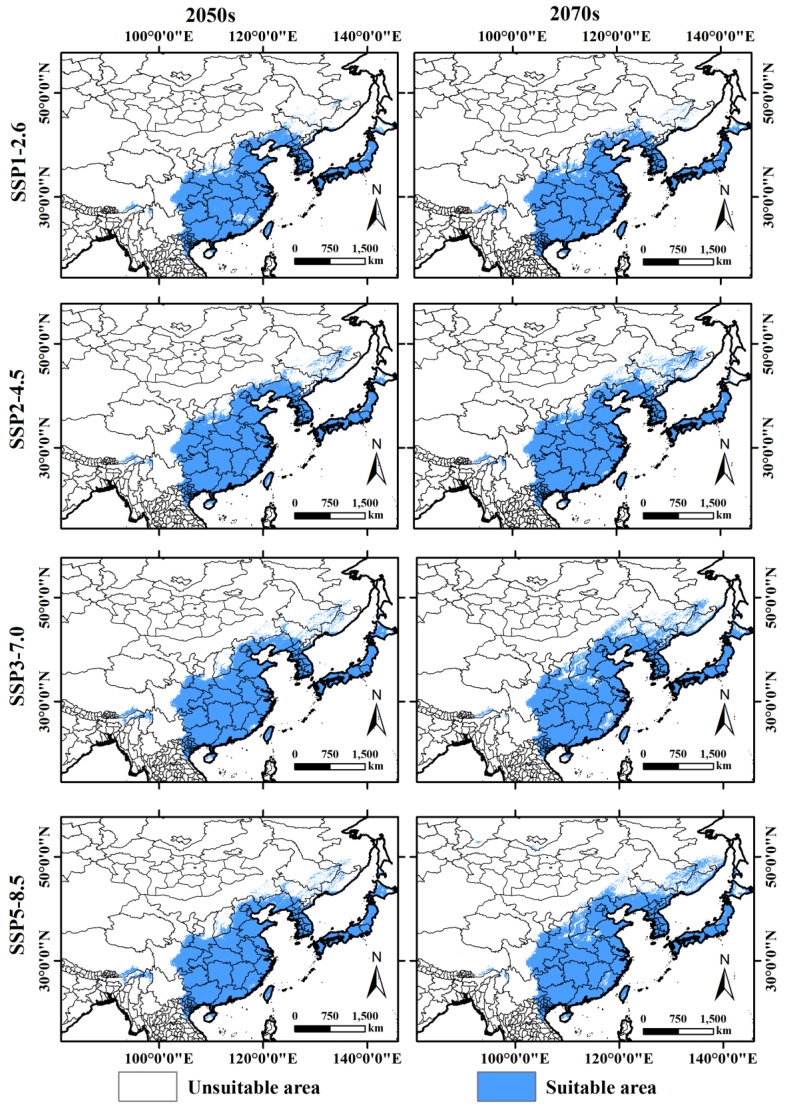
Potential future distribution areas of *A. rugicollis* under different climate models in the future.

**Figure 8 insects-15-00930-f008:**
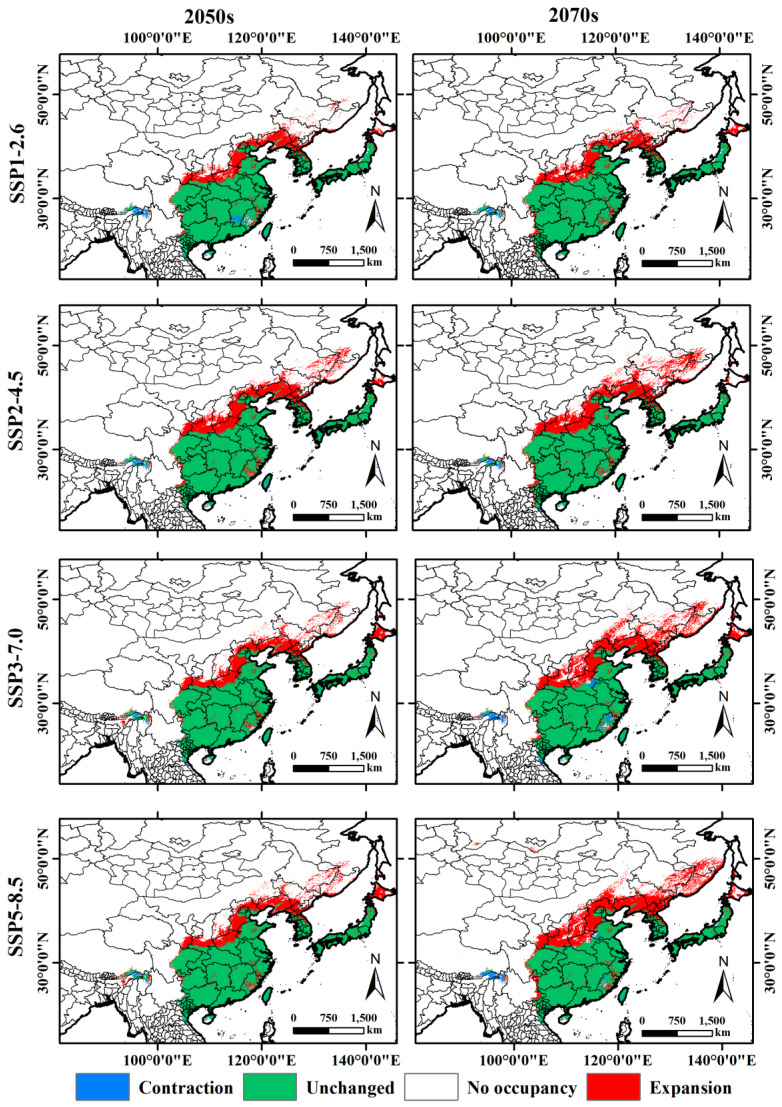
Region of relative change in *A. rugicollis* under different future climate scenarios.

**Figure 9 insects-15-00930-f009:**
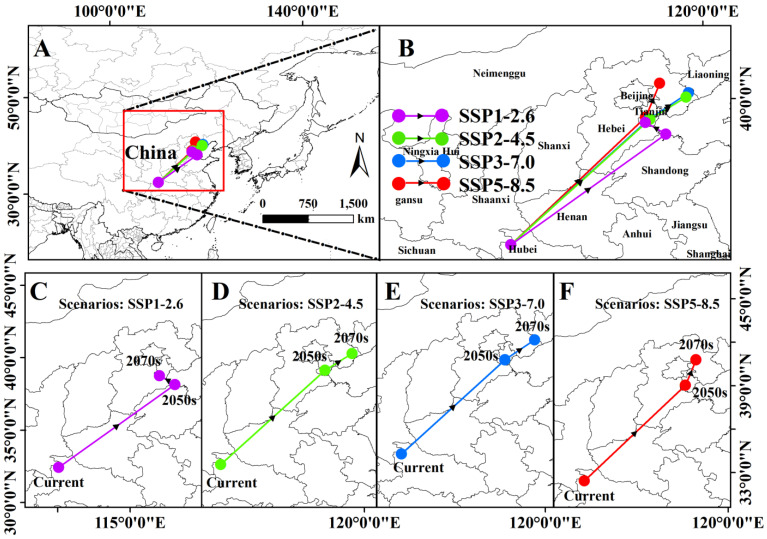
(**A**) Spatial trajectory of *A. rugicollis* under future climate scenarios; (**B**) spatial change trajectories of *A. rugicollis* under four different future climate scenarios; (**C**) SSP1-2.6; (**D**) SSP2-4.5; (**E**) SSP3-7.0; (**F**) SSP5-8.5.

**Table 1 insects-15-00930-t001:** Environmental variables and screening results based on Pearson’s correlation of |*r*| > 0.7 and VIF > 10.

Abbreviation	Environmental Variables	Operation (|*r*| > 0.7)
Bio1	Annual mean temperature (°C)	Eliminate
Bio2	Mean diurnal range (°C)	Eliminate
Bio3	Isothermality (BIO2/BIO7) (×100)	Eliminate
Bio4	Temperature seasonality (standard deviation ×100)	Eliminate
Bio5	Maximum temp of warmest month (°C)	Eliminate
Bio6	Minimum temp of coldest month (°C)	Eliminate
Bio7	Temperature annual range (°C)	Eliminate
Bio8	Mean temp of wettest quarter (°C)	Eliminate
Bio9	Mean temp of driest quarter (°C)	Eliminate
Bio10	Mean temp of warmest quarter (°C)	Eliminate
Bio11	Mean temp of coldest quarter (°C)	Eliminate
Bio12	Annual precipitation (mm)	Eliminate
Bio13	Precipitation of wettest month (mm)	Eliminate
Bio14	Precipitation of driest month (mm)	Eliminate
Bio15	Precipitation seasonality (mm) (Coefficient of Variation)	Retain
Bio16	Precipitation of wettest quarter (mm)	Eliminate
Bio17	Precipitation of driest quarter (mm)	Eliminate
Bio18	Precipitation of warmest quarter (mm)	Eliminate
Bio19	Precipitation of coldest quarter (mm)	Retain
Bio20	Elevation (m)	Retain
Bio21	Slope	Retain
Bio22	Aspect	Retain
Bio23	Annual_mean_UV-B	Eliminate
Bio24	UV-B_seasonality	Retain
Bio25	Mean_UV-B_of_highest_month	Eliminate
Bio26	Mean_UV-B_of_lowest_month	Eliminate
Bio27	Sum_of_UV-B_radiation_of_highest_quarter	Eliminate
Bio28	Sum_of_UV-B_radiation_of_lowest_quarter	Eliminate
Bio29	Global human footprint	Retain
Bio30	Global human influence index	Eliminate
Bio31	Population density	Retain

**Table 2 insects-15-00930-t002:** Accuracy evaluation of ensemble model.

Shared Socioeconomic Pathways	Sensitivity	Specificity	TSS	ROC
Natural environmental	0.95	0.94	0.89	0.99
Natural environmental + Human activity	0.94	0.93	0.87	0.98

**Table 3 insects-15-00930-t003:** Predicted suitable areas for *A. rugicolli* under current and future climatic conditions.

Shared Socioeconomic Pathways	Predicted Area (×10^4^ km^2^)		Comparison with Current Environmental Variables Distribution (%)	
	Unsuitable Area	Suitable Area	Unsuitable Area	Suitable Area
Current natural environment	5138.46	255.58	-	-
Current natural environment + human activity	5150.30	243.75	0.23	−4.63
Future-SSP1-2.6 2041–2060	5074.11	319.93	−1.25	25.18
Future-SSP1-2.6 2061–2080	5061.95	332.09	−1.49	29.94
Future-SSP2-4.5 2041–2060	5042.33	351.72	−1.87	37.61
Future-SSP2-4.5 2061–2080	5028.42	365.62	−2.14	43.06
Future-SSP3-7.0 2041–2060	5049.98	344.06	−1.72	34.62
Future-SSP3-7.0 2061–2080	5021.82	372.22	−2.27	45.64
Future-SSP5-8.5 2041–2060	5049.22	344.82	−1.74	34.92
Future-SSP5-8.5 2061–2080	5003.94	390.10	−2.62	52.63

**Table 4 insects-15-00930-t004:** Relative area of change in *A. rugicollis* under different future climate scenarios.

Shared Socioeconomic Pathways	Predicted Area (×10^4^ km^2^)			
	Contraction	Unchanged	No Occupancy	Expansion
Future-SSP1.0-2.6 2041–2060	7.33	5066.78	248.25	71.68
Future-SSP1.0-2.6 2061–2080	4.79	5057.16	250.79	81.31
Future-SSP2.0-4.5 2041–2060	4.30	5038.03	251.28	100.43
Future-SSP2.0-4.5 2061–2080	4.39	5024.03	251.19	114.43
Future-SSP3.0-7.0 2041–2060	3.33	5046.65	252.25	91.82
Future-SSP3.0-7.0 2061–2080	9.75	5012.08	245.83	126.39
Future-SSP5.0-8.5 2041–2060	3.15	5046.07	252.43	92.39
Future-SSP5.0-8.5 2061–2080	6.26	4997.69	249.32	140.78

**Table 5 insects-15-00930-t005:** Latitude, longitude, and distance of migratory changes in potential distribution centers of *A. rugicollis* under different future climate scenarios.

Shared Socioeconomic Pathways	Longitude (°E)	Latitude (°N)	Center Migration Distance (km)
Current	110.06	32.43	-
Future-SSP1.0-2.6 2041–2060	118.11	38.16	968.50
Future-SSP1.0-2.6 2061–2080	117.04	38.77	944.87
Future-SSP2.0-4.5 2041–2060	117.28	38.94	972.89
Future-SSP2.0-4.5 2061–2080	119.15	40.09	1176.82
Future-SSP3.0-7.0 2041–2060	117.23	38.94	969.59
Future-SSP3.0-7.0 2061–2080	119.28	40.31	1201.98
Future-SSP5.0-8.5 2041–2060	117.05	39.04	967.33
Future-SSP5.0-8.5 2061–2080	117.79	40.80	1156.48

## Data Availability

Data in this study are available from the corresponding author.

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
