# Peer review of "Assessment of the Potential Suitable Habitat of Apriona rugicollis Chevrolat, 1852 (Coleoptera: Cerambycidae) Under Climate Change and Human Activities Based on the Biomod2 Ensemble Model"

_insects, 2024, doi:10.3390/insects15120930_

Round 1
Reviewer 1 Report
Comments and Suggestions for Authors
This paper presents species distribution modeling efforts to predict where the distribution of a pest of tree species may expand over time. I thought this paper was well written. I particularly appreciated the effort the authors took to present the details of the modeling techniques, data assembly and analyses, and model testing. I feel the methods are particularly well developed, and the results and discussion accurately and reasonably interpret the spatial models. I only have a few comments, which I have made in the body of the manuscript using comment tools. My comments mainly deal with small editorial changes/suggestions, as well as suggest clearer wording of some phrases or sentences. One final comment is that the authors should consider expanding the table and figure titles to more fully describe what is presented in each. While the figures and tables are mostly intuitive, more explanation of the values, coefficients, parameters, etc. presented in each would enhance the experience of the reader and make interpreting the tables and figures easier. I feel that if the authors address my comments this paper will be ready for publication.

Author Response
Dear Mr. Michael Wang and Reviewers:
Thank you for your letter and for the reviewers' comments concerning our manuscript entitled “Assessment the Potential Suitable Habitat of Apriona rugicollis Chevrolat, 1852 (Coleoptera: Cerambycidae) under Climate Change and Human Activities Based on Biomod2 Ensemble Model” (ID: insects-3320970). We appreciate the detailed and constructive comments provided by the reviewers. Those comments are all valuable and very helpful for revising and improving our paper, as well as the important guiding significance to our research. We have studied comments carefully and have made corrections which we hope meet with approval.
We hope this revised manuscript addresses your concerns and look forward to hearing from you.
With regards,
Yours faithfully,
Ping Wang,
Wenkai Wang,
Yangtze University
Jingzhou, 434000, P. R. China
E-mail: wangping1992@yangtzeu.edu.cn; wwk@yangtzeu.edu.cn
Reply to Reviewer #1:
Dear Reviewer,
Thank you very much for taking the time to review the manuscript carefully and for your very constructive comments. We also appreciate your clear and detailed feedback and hope the explanation adequately addresses all your questions. We have done our best to improve the manuscript and have made some changes to the manuscript. These changes do not affect the content or framework of the paper. To facilitate discussion, we have first retyped your comments in italic bold and then listed our response to each comment below. We sincerely thank the reviewers for their enthusiastic work and hope that these revisions will be acknowledged.
Thank you again for your comments and suggestions.
Comment 1: I suggest using the term 'movement' or 'shift' here and reserve the term 'migration' for animal or insects species.
Author's response:
Thank you for your valuable suggestions. We have revised the relevant content in accordance with your suggestions in order to make the presentation more accurate and clearer (Line 47).
Comment 2: Use 'migrate' here.
Author's response:
Thank you for your suggestion. In line with your comments, we have amended the expression to read “migrate”. Thank you for your help so that we can further improve the manuscript (Line 48)。
Comment 3: What is this? Does this mean changes in the distribution over time, or how spatial/climate changes influence species evolution somehow, or something else? Consider more descriptive terminology.
Author's response:
Thank you for your interest in terminological clarity. To avoid ambiguity, we have adjusted the terminology to “distribution dynamics” to more accurately convey this concept, based on your suggestion.
Thank you for your valuable suggestions to improve the clarity of our presentation (Line 110).
Comment 4: I don't think that web links should be listed in the body of the paper. Each of these links should be a reference and formatted according to journal guidelines.
Also, the numbering scheme listing your sources is not necessary. Once you remove the link and cite your data sources normally, just use commas to list the sources.
Author's response:
Thank you for your comment. We have carefully checked the journal's submission requirements as well as recent articles published in Insects and found that the way we have cited the web links in the body of the article is in line with the journal's requirements. The purpose of this is to increase the transparency and accessibility of the data sources and to facilitate access and validation by other researchers.
If the editorial board suggests further adjustments to this during the final review stage, we will actively cooperate to modify the format.
Comment 5: data from field survevs conducted by researchers from 2013 to present in vanous provinces in China
Author's response:
We have amended the relevant content in response to your comments to present the data sources more clearly (Line 124-125).
Comment 6: I suggest also presenting the resolution for these data as kilometers in parentheses at the first mention of this particular resolution, just so it is more intuitive for the reader. You could use the median latitude of your extent for the conversion.
Author's response:
Thank you for your suggestion. In order to improve the visualization of the resolution of the data, we have added the unit “4.6 km” in parentheses when the resolution is first mentioned, which will help the reader to understand the resolution of the data more easily (Line 144).
Comment 7: As stated above, websites are to be cited in the body of the text the same as publications (i.e., with a number), and the citation for the website listed in the References section. This is the example formate from the Insects Instructions for Authors found online for citing websites:
Websites:
- Title of Site. Available online: URL (accessed on Day Month Year).
Author's response:
Thank you for your suggestion. We have re-downloaded and carefully reviewed the journal's latest submission formatting instructions as well as recent articles published in Insects and found that our approach of directly citing web links in the body of the text is in line with the journal's requirements. This treatment is intended to increase the transparency and accessibility of the data sources for other researchers to review and validate.
If the editorial board suggests further adjustments to this citation method during the final review stage, we will actively work with them to modify the format.
Thank you again for raising this issue and helping us to further improve the citation specification of the article.
Comment 8: Was there a value threshold for VIF?
Author's response:
Thank you for your valuable comments. In our study, we adopted the widely recognised Variance Inflation Factor (VIF) threshold, which means that severe multicollinearity may exist when the VIF value is greater than 10. Therefore, we gradually removed the variables with VIF values greater than 10 until all retained variables had VIF values less than 10.This approach ensured the stability of the model and the accuracy of the prediction results, and avoided the potential impact of multicollinearity on the performance of the model (Line 177).
Comment 9: Reword this sentence. Current wording suggests that eight variables were eliminated, yet the opposite is true. I suggest: 'Finally, eight environmental factors were selected that were relatively independent and........'
Author's response:
Thank you for your valuable comments. Based on your suggestion, we have revised the relevant expression to convey the information more clearly. The original sentence “eight variables were eliminated” was misleading and has been changed to “eight environmental factors were selected”, which more accurately reflects our screening process. Thank you for your careful review, which helped us to improve the accuracy of the paper (Line 178).
Comment 10: I suggest rewording the Table title to something like: 'Table 1. Environmental variables and screening results based on Pearson's correlation of |r|>0.7 and value inflation factors x>0.XX.'
Author's response:
Thank you for your valuable suggestion. We have modified the title of the table according to your suggestion and it now reads: “Environmental variables and screening results based on Pearson’s correlation of |r|>0.7 and VIF > 10”. This change makes the title clearer and accurately conveys the screening criteria. Thank you again for your help (Table 1, Line 184-185).
Comment 10: Define the terms AUC and TSS here instead of the next section, since this is the first mention of these terms.
Author's response:
Thank you for your valuable suggestions. We will revise the relevant content, which will help the reader to understand the terms more clearly (Line 197-198).
Comment 12: I suggest rewording this sentence, so that the relation of specificity and sensitivity to the different error types is more clear. For exampe: The TSS value is a statistic based on the confusion matrix that combines the sensitivity (quantification of omission error) and specificity (quantification of commission error) of the model.
Author's response:
Thank you for your valuable suggestions. We have modified the sentence based on your comment to make the relationship between specificity, sensitivity and different error types clearer. The modified sentence is as follows: “The TSS value is a statistic based on the confusion matrix that combines the sensitivity (quantification of omission error) and specificity (quantification of commission error) of the model.” (Line 219-221).
Comment 13: Is this the same as 'spatial evolution pattern' mentioned in objective 4? If so, then the terminology needs to be consistent with your objective and vise versa. So change wording of either this heading or the wording of the objective to match more closely.
Author's response:
Thank you for your careful review. There is indeed some terminological inconsistency between your reference to “Spatial evolution pattern” and the formulation in the fourth objective. In order to maintain terminological consistency, we will change it to “Spatial distribution dynamics” to ensure a better match between the objective and the title (Line 249).
Comment 15: I suggest including more information in your Figure titles. For instance, this title could include the ranges of p-values for TSS and ROC.
Additionally, at it's current size, the p-values are virtually illegible. I suggest requesting of the editor to have the type setter/publisher increase the figure size to it's maximum extent possible on the page, so that the values are able to be read. To accomplish this, you may have rearrange your model legend and/or crop in on your original image to allow for increases in size on the page.
Author's response:
Thank you for your valuable suggestions.TSS and ROC values are two metrics used to assess the performance of a model, and they are different in nature and calculation, so a direct comparison of their p-values is not appropriate. In addition, we will communicate with the editors during the layout stage of the article to request for appropriate resizing of the charts to ensure that the values are clear and easy to read. If necessary, we will also adjust the layout of the model legends or crop the original images to optimise the use of page space.
Thank you again for your valuable comments, which will help to further enhance the presentation of our work.
Comment 16: As in Figure 4, I suggest elaborating on this figure title. You could list what each of teh bioclim variables on the x-axis are, so that the reader doesn't have to jump back and forth from the figure to the text to interpret the figure.
Author's response:
Thank you for your suggestions. We will further enrich the chart titles by explicitly listing each bioclimatic variable on the x-axis so that readers can more easily interpret the charts without having to jump back and forth between the charts and the text (Figure 4).
Comment 17: Instead of this sentence, which is a restatement of the previous sentence, could you include a clarifying sentence based on the maps presented in Figure 6? Something like: 'Reductions in range occurred in areas with.....' (higher human development or greater deforestation or etc.).
Author's response:
Thank you for your suggestion. In order to avoid redundancy, we have deleted the last sentence in the original text and further clarified the specific areas of decrease based on the map in Figure 6. For example, “the areas of decrease were mainly located in Kachin State, Myanmar, Guangxi, Guangdong, Fujian, Jiangxi, Hunan, Hubei, Shandong and Henan, China.” (Line 327-329).
Comment 18: Move table so that it is all on one page.
Author's response:
Thank you for your suggestions. We will adjust the layout of the table to ensure that it appears exactly on the same page to improve readability and clarity (Table 3).
Comment 19: I don't think the elevational shading is necessary on this map. I suggest removing it, but keep all the political boundaries. This change would make the changes in centers of distribution and accompanying text more legible for the reader.
Author's response:
Thank you for your valuable suggestions. We will remove the elevation layer from the map while keeping all political boundaries. This will make the changes to the distribution centres and the associated text clearer and easier to read (Figure 9).
Comment 20: This is a long paragraph, and it discusses possible reasons the different variables may help or hinder distributional changes. I suggest breaking down the discussion of each variable into different paragraphs, so that it is clear which variable is being discussed.
Author's response:
Thank you for your suggestion. We will split this paragraph into multiple paragraphs discussing the role and impact of each variable separately, which will give the reader a clearer understanding of the different impacts of each variable on changes in the distribution (Section 4.2).
Comment 21: I suggest using different qualifiers to start one of these sentences. Generally, if 'on the one hand' and 'on the other hand' are used to contrast points. However, both items presented in these two sentences are harmful to insects. So, I suggest using something like 'Additionally' or 'Furthermore' to start the second of these sentences.
Author's response:
Thank you for your suggestion. We have changed the beginning of the sentence in response to your suggestion, avoiding the use of the words “on the one hand” and “on the other hand” to contrast two factors that are unfavourable to insects. Sentences now begin with “Additionally” and “Furthermore” to make the meaning clearer (Line 458 and Line 460).
We would like to take this opportunity to thank you for all your time involved and this great opportunity for us to improve the manuscript. We hope you will find this revised version satisfactory.
Your Sincerely,
Ping Wang,
Wenkai Wang,
End of Reply to Reviewer #1
Reviewer 2 Report
Comments and Suggestions for Authors
All comments are attached

Author Response
Dear Mr. Michael Wang and Reviewers:
Thank you for your letter and for the reviewers' comments concerning our manuscript entitled “Assessment the Potential Suitable Habitat of Apriona rugicollis Chevrolat, 1852 (Coleoptera: Cerambycidae) under Climate Change and Human Activities Based on Biomod2 Ensemble Model” (ID: insects-3320970). We appreciate the detailed and constructive comments provided by the reviewers. Those comments are all valuable and very helpful for revising and improving our paper, as well as the important guiding significance to our research. We have studied comments carefully and have made corrections which we hope meet with approval.
We hope this revised manuscript addresses your concerns and look forward to hearing from you.
With regards,
Yours faithfully,
Ping Wang,
Wenkai Wang,
Yangtze University
Jingzhou, 434000, P. R. China
E-mail: wangping1992@yangtzeu.edu.cn; wwk@yangtzeu.edu.cn
Reply to Reviewer #2:
Dear Reviewer,
Thank you very much for taking the time out of your busy schedule to review my manuscript and for your very encouraging comments. We also appreciate your clear and detailed feedback and hope that these explanations have adequately addressed all of your questions. In the remainder of this letter, we will discuss each of your comments and our corresponding responses separately. To facilitate this discussion, we have retyped your comments in italicized bold and set out our responses below.
Comment 1: Since one of the research objectives of the author's paper is to compare differences in the extent and area of the distribution of A. rugicollis habitat in the current period under scenarios with and without anthropogenic activities, I suggest that the author should talk about the impact of human activities on the distribution of pests in the Introduction section, and why human activity variables should be included in the prediction of A. rugicollis? In other words, has the existing research documented that human activities affect the distribution of A. rugicollis? If not? What is the purpose of adding human activities variables?
Author's response:
Thank you for your comments and suggestions. Your suggestions were very helpful and we have revised the introduction section based on your feedback. We have added information about the effects of human activities on pest distribution to the introduction. At the same time, we will further elaborate on why we included the human activity variable in the prediction model. Although there are relatively few studies on the relationship between human activities and the distribution of A. rugicollis in the existing literature, it has been shown that human activities may play an important role in altering habitat structure and function, disrupting ecosystem balance, and reducing species diversity. Therefore, when predicting the distribution of A. rugicollis, we believe that considering anthropogenic variables can help to assess potential changes in its distribution more comprehensively. We will explain this in detail in the introduction to the newly revised manuscript to ensure that the reader understands the rationale and necessity of including these variables (Line 51-56).
Comment 2: Line 100-110: The author collected species distribution data from different data sources, but I have a few questions. First, during the collection process, did the author only collect distribution data with longitude and latitude? If not, how did they obtain the longitude and latitude data for the distribution areas without longitude and latitude data? Second, what is the spatial resolution of the distribution area? County level or other levels? Third, the author obtained some distribution data from the GBIF and iNaturalist, but I did not see some problematic data deleted. Generally speaking, the data on the GBIF and iNaturalist cannot be used directly but needs further cleaning. I hope the author will solve these problems.
Author's response:
Thank you for your questions and valuable comments. During the collection of species distribution data, we mainly prioritised the collection of records with clear latitude and longitude information. For records lacking specific latitude and longitude information, we obtained the geographic centroids of the corresponding administrative divisions as the approximate latitude and longitude coordinates based on their geographic descriptions (e.g., location at the city level) via Google Earth software, and normalized these data to ensure the accuracy of spatial positioning and reliability of the data as much as possible. Meanwhile, regarding the spatial resolution of the data, most of the species distribution records we collected were based on the locality level, which generally reached the local or county level of accuracy to ensure the reliability of the analyses. In addition, data from platforms such as GBIF and iNaturalist usually require further cleaning before they can be used for research. For this reason, we took rigorous screening and cleaning measures during data processing. We first deleted duplicate records, and then sparsified the data (at 4.6 km intervals) using the “spThin” package to reduce the impact of spatial autocorrelation on the results (Line 125-128 and Line 131-138).
Comment 3: Line 203-215: The author has built three models here, but I have a question. According to my understanding, the model 3 built by the author is used to predict the future distribution area, but in fact this result is obtained on the basis of model 1. In other words, your model 3 is model 1. If not, you directly built a model based on the future environmental data and the data of the distribution points to predict the future distribution area, but this is obviously wrong. Therefore, I hope the author will carefully revise this part.
Author's response:
Thank you for your question and valuable comments. Your understanding is correct in that Model 3 is actually a further analysis based on the predictions of Model 1. Model 1 was used to construct a predictive model of species suitability based on current environmental data and distribution points, whereas Model 3 builds on Model 1 by incorporating future environmental data to predict the likely distribution areas of species in the future. Thus, Model 3 was not reconstructed directly from scratch based on future environmental data and distribution points. In order to avoid misunderstanding, we will elaborate this part in more detail and clearly in the revised version, emphasising that Model 3 is a predictive analysis using future environmental data under the framework of Model 1 (Line 233-236).
Comment 4: First of all. The part “4.1. Reliability and Accuracy of Ensemble model”. This paragraph illustrated the advantages of the ensemble model and shows that the research results are more reliable. However, it is inappropriate to put it here because the advantages of the ensemble model have been shown in the introduction part. Therefore, I suggest deleting this part. Second, the discussion section is mostly a repetition of the results of this paper, so I suggest that the author carefully revise the discussion section and present it to the readers in a clear and concise form.
Author's response:
Thank you for your feedback and suggestions. We agree that the discussion section should avoid repeating what is in the introduction. However, we believe that section 4.1 is important for validating the reliability and scientific value of the results in this paper. In order to avoid redundancy with the introduction or the results section, we will modify Section 4.1. While the Introduction section focuses on the theoretical aspects of the advantages of the integrated model, Section 4.1 further validates the reliability of our results with specific data and metrics of model accuracy (e.g., TSS and AUC values). We have noted that the discussion section does have the problem of repeating the results. For this reason, we will remove descriptions that are too lengthy or repetitive with the results section and keep the key information (Section 4.1).
Comment 5: Line 56-58: “SDMs simulate.......... occurrence records and environmental variables”. This sentence is not appropriate. The author's sentence only expresses the correlative speciesspecies distribution model, so I suggest that the front of this sentence should be revised to: Correlative species distribution models........
Author's response:
Thank you for your feedback. We have revised the sentence to make it clearer and more accurate based on your suggestion. The revised sentence is as follows: “Correlative species distribution models (SDMs) simulate suitable habitats for species by combining statistical and machine learning algorithms using their occurrence records and environmental variables, and are irreplaceable in assessing the potential impacts of climate change on species distribution.” (Line 64).
Comment 6: Line 73-82: Here the author is expressing the harm of pests to trees, which is necessary. However, the author does not explain how much damage the pests have caused, such as the annual economic losses or the area of trees damaged. Adding such a sentence can better illustrate the harmfulness of pests.
Author's response:
Thank you for the reminder. As there is currently no literature that clearly identifies the specific economic losses caused by Apriona rugicollis, we will nonetheless improve this section by using a more careful formulation that emphasises its potential harm and impact (Line 91-97).
Comment 7: Line 120-122: deleted “This approach effectively................. spatial distribution prediction”
Author's response:
Thank you for your suggestion. We have removed the section “This approach effectively reduced duplicate records and improved the scientific validity and reliability of the model, providing a solid data base for subsequent spatial distribution prediction.”. This modification makes the paragraph more concise and clearer (Section 2.1).
Comment 8: Line 111: Please state the source of the vector map used in the article in Figure 1
Author's response:
Thank you for your suggestion. We will clearly state the source of the vector maps in Figure 1 in the main text, which will help to increase the transparency of the data sources and make it easier for other researchers to refer to them (Line 161-164).
Comment 9: For ease of reading, the names of the regions where the subsequent model predicts the presence of suitable areas can be added in Figure 1.
Author's response:
Thank you for your suggestion. We will modify Figure 1 by adding appropriate place names so that the reader can visualize it better (Line 140).
Comment 10: Please indicate the time of future environmental change
Author's response:
Thank you for your suggestions. We will add specific time frames for future environmental change. In this paper, we project future environmental changes for the 2050s (2041-2060) and 2070s (2061-2080) (Line 149).
Comment 11: Line 137-138: What coordinate system is used when calculating slope and aspect using elevation data in ArcGIS?
Author's response:
Thank you for your question. We use the WGS 1984 coordinate system when calculating slope and aspect in ArcGIS. This coordinate system ensures that terrain features are accurate on a global scale (Line 156).
Comment 12: Line 154-155: The author uses pictures to show the results of the correlation analysis, but this method makes the correlation coefficient values in the pictures a bit small, which is not very clear, and also lacks significance level values. I suggest that the author use a table to show the results.
Author's response:
Thank you for your valuable suggestions. Due to the large number of environmental factors involved in our study (31 in total), we believe that including all of them in the table may result in insufficient chart space. We believe that by presenting Figure 2 and Table 1, we have been able to present the correlation information clearly and effectively support the results of our analyses. In addition, we will be uploading the tables from the correlation analyses for your review.
Thank you again for your understanding and suggestions. If you have any further questions or suggestions, please feel free to communicate with us.
Comment 13: Line 157-158: “had a large impact on model predictions”? VIF and correlation distribution cannot shown your above conclusion.
Author's response:
Thank you for your comments. We accept your suggestion and will revisit the results of the VIF and correlation analyses to ensure that our conclusions are more in line with the data support. The VIF and correlation analyses are primarily used to assess multicollinearity and do not directly affect model predictions. Therefore, we will revise this statement to avoid any misunderstanding (Line 177-178).
Comment 14: Please increase the units of all environment variables in Table 1
Author's response:
Thank you for your suggestion. We have added the units for all environment variables in Table 1 to ensure clarity and ease of reference.
Comment 15: Line 186-202: This section introduces AUC and TSS, but it is recommended that this part can be integrated into section 2.3
Author's response:
Thank you for your comments and suggestions. We have revised the content to enhance the coherence of the methodology section and to reduce repetitive content (Line 197-198 and Line 219-222).
Comment 16: Line 218: ranging from 0 to 1000?Please confirm
Author's response:
Thanks for your comment. This is because the biomod2 package by default scales the predictions by a factor of 1000 to improve accuracy, so the output range of the model is usually 0-1000, not 0-1.
Comment 17: Line 230: No occupancy refers no distribution of the species under the new scenario? As far as I know, it refers to places where there is no distribution in both now and in the future. Please confirm further. In addition, No occupancy does not reflect changes in suitable areas.
Author's response:
Thank you for your comments. We agree with you that ‘no occupancy’ usually refers to areas where a species has no distribution in both current and future scenarios. We apologise for the confusion caused by our use of this term. We will make it clear in the new manuscript that ‘no occupancy’ refers to areas where species are not distributed in both current and future scenarios. We will also more clearly distinguish between ‘no occupancy’ and changes in suitable areas, since ‘no occupancy’ does not directly reflect changes in suitable habitat (Line 255-256).
Comment 18: Line 238: predicting future ecosystem dynamics?I don't understand the meaning of this sentence
Author's response:
Thank you for your question. We will rephrase the sentence to convey its meaning more clearly. In the new manuscript we will change it to “This analysis not only reveals the response patterns of species to climate change, but also provides an important spatial reference for predicting future changes in the spatial distribution of species.” (Line 262-264)
Comment 19: Please add scale bars and compasses to Figures 6 to 9
Author's response:
Thank you for your suggestion. We will add a scale and a compass to Figures 6 to 9 so that the reader can better understand the spatial information in the figures, and the addition of these elements will help to improve the readability and intuition of the graphics (Figures 6 to 9).
Comment 20: Is it 2040-2060 or 2041-2060? Please check other time periods as well in Table 3
Author's response:
Thank you for your suggestion. We have corrected “2040-2060” to “2041-2060” and have also checked and amended the other time periods for correctness (Table 3 to5).
Comment 21: Line 312: 2050 and 2070?
Author's response:
Thank you for your comment. We have amended “2050 and 2070” to “2050s and 2070s” (Line 339).
Comment 22: Line 381-388:delete“A. rugicollis................with A. rugicollis are also reduced”
Author's response:
Thank you for your suggestion. Based on your comments, we have removed the relevant content (Line 412).
Comment 23: Line 422-425: delete “In this study.................human activities”
Author's response:
Thank you for your suggestion. Based on your comments, we have removed the relevant content (Line 443).
Thank you again for your attention. Looking forward to your reply.
Your sincerity,
Ping Wang,
Wenkai Wang,
End of Reply to Reviewer #2
Round 2
Reviewer 2 Report
Comments and Suggestions for Authors
Comments are attached

Author Response
Dear Mr. Michael Wang and Reviewers:
Thank you for your letter and for the reviewers' comments concerning our manuscript entitled “Assessment the Potential Suitable Habitat of Apriona rugicollis Chevrolat, 1852 (Coleoptera: Cerambycidae) under Climate Change and Human Activities Based on Biomod2 Ensemble Model” (ID: insects-3320970). We appreciate the detailed and constructive comments provided by the reviewers. Those comments are all valuable and very helpful for revising and improving our paper, as well as the important guiding significance to our research. We have studied comments carefully and have made corrections which we hope meet with approval.
We hope this revised manuscript addresses your concerns and look forward to hearing from you.
With regards,
Yours faithfully,
Ping Wang,
Wenkai Wang,
Yangtze University
Jingzhou, 434000, P. R. China
E-mail: wangping1992@yangtzeu.edu.cn; wwk@yangtzeu.edu.cn
Reply to Reviewer #2:
Dear Reviewer,
Thank you very much for taking the time out of your busy schedule to review my manuscript and for your very encouraging comments. We also appreciate your clear and detailed feedback and hope that these explanations have adequately addressed all of your questions. In the remainder of this letter, we will discuss each of your comments and our corresponding responses separately. To facilitate this discussion, we have retyped your comments in italicized bold and set out our responses below.
Comment 1: Line 227-240: Since Model 3 is based on Model 1, I think the term "three different ensemble models" should not be used in the description. As I said originally, you only built 2 models, so I hope the author can further revise this part carefully to make it more logical and clear.
Author's response:
Thank you for your comments. We recognize that the expression “3 different ensemble models” can be confusing, as Model 3 is indeed based on Model 1. In response to your suggestion, we have revised the relevant section to describe the model building process in a clearer and more logical way (Line 229-242).
Comment 2: Line 429-431:TSS value have the classification ability in distinguish between suitable and unsuitable area? How to understand this sentence
Author's response:
Thank you for your comment, we recognize that there may have been ambiguity in this statement, which may have led to confusion. The TSS value itself does not have classification ability, but is a measure of the performance of the classification model. The closer the TSS value is to 1, the better the model's classification performance is, and the more accurately it can distinguish between suitable and unsuitable areas. We have adjusted the relevant expressions to more accurately reflect the meaning of the TSS value (Line 431-434).
Comment 3: Line 94-95:“the area of forest damaged by the feeding behaviour of A. rugicollis continues to expand”. Please add references
Author's response:
Thank you for your comments. We have added relevant references to support this statement.
The references are listed below:
- Apriona rugicollis. EPPO datasheets on pests recommended for regulation. EPPO, 2024.
- Ji, L.; Wang, Z.; Wang, X.; An, L. Forest Insect Pest Management and Forest Management in China: An Overview. Environ Manage. 2011, 48, 1107–
Comment 4: The author only names Japan in Figure 1, and the names of other countries are not described. So I suggest adding this section.
Author's response:
Thank you very much for your careful review of the contents of the figure notes. Due to our inadvertence, only Japan’s was labelled in the figure caption of Figure 1, which may cause disturbance to the readers' understanding. We have revised the figure caption according to your suggestion by adding the names of other countries to ensure that the content is more complete and clear (Line 140-141).
Thank you again for your valuable comments!
Comment 5: Line 149:“For future climate data (2050s and 2070s) ”modify as For future climate data from 2041-2060 (2050s) and 2061-2080 (2070s)
Author's response:
Thank you for your suggestion, we have revised the relevant section based on your comments to more accurately express the timeframe (Line 150).
Comment 6: Line 196-226:My opinion is to combine them. In the first paragraph, you introduced AUC and TSS, and in the second paragraph, you said that they are indicators for evaluating models. You can write this part directly into the first paragraph. Your second paragraph and the first paragraph are both telling the same story.
Author's response:
Thank you for your suggestions on the logical structure of the section. We fully understand your comments and, in accordance with your suggestions, we have integrated and streamlined the two paragraphs to ensure coherence, avoid repetition and make the article more concise and clear (Subsection 2.3).
再次感谢您的关注。期待您的回复。
您的真诚,
王平,
王文凯,
回复审阅者 #2 结束
Round 3
Reviewer 2 Report
Comments and Suggestions for Authors
Dear author, thank you very much for taking my opinion, now I have no more comments